# Off-Policy Imitation Learning from Observations

**Zhuangdi Zhu**
Michigan State University
`zhuzhuan@msu.edu`

**Kaixiang Lin**
Michigan State University
`linkaixi@msu.edu`

**Bo Dai**
Google Research
`bodai@google.com`

**Jiayu Zhou**
Michigan State University
`jiayuz@msu.edu`

## Abstract

Learning from Observations (LfO) is a practical reinforcement learning scenario from which many applications can benefit through the reuse of incomplete resources. Compared to conventional imitation learning (IL), LfO is more challenging because of the lack of expert *action* guidance. In both conventional IL and LfO, *distribution matching* is at the heart of their foundation. Traditional distribution matching approaches are sample-costly which depend on *on-policy* transitions for policy learning. Towards sample-efficiency, some *off-policy* solutions have been proposed, which, however, either lack comprehensive theoretical justifications or depend on the guidance of expert actions. In this work, we propose a sample-efficient LfO approach which enables *off-policy* optimization in a *principled* manner. To further accelerate the learning procedure, we regulate the policy update with an inverse action model, which assists distribution matching from the perspective of *mode-covering*. Extensive empirical results on challenging locomotion tasks indicate that our approach is comparable with state-of-the-art in terms of both sample-efficiency and asymptotic performance.

## 1 Introduction

Imitation Learning (IL) has been widely studied in the reinforcement learning (RL) domain to assist in learning complex tasks by leveraging the experience from expertise [1, 2, 3, 4, 5]. Unlike conventional RL that depends on environment reward feedbacks, IL can purely learn from expert guidance, and is therefore crucial for realizing robotic intelligence in practical applications, where demonstrations are usually easier to access than a delicate reward function [6, 7].

Classical IL, or more concretely, Learning from Demonstrations (LfD), assumes that both *states* and *actions* are available as expert demonstrations [8, 2, 3]. Although expert actions can benefit IL by providing elaborated guidance, requiring such information for IL may not always accord with the real-world. Actually, collecting demonstrated actions can sometimes be costly or impractical, whereas observations without actions are more accessible resources, such as camera or sensory logs. Consequently, Learning from Observations (LfO) has been proposed to address the scenario without expert actions [9, 10, 11]. On one hand, LfO is more challenging compared with conventional IL, due to missing finer-grained guidance from actions. On the other hand, LfO is a more practical setting for IL, not only because it capitalizes previously unusable resources, but also because it reveals the potential to realize advanced artificial intelligence. In fact, learning without action guidance is an inherent ability for human being. For instance, a novice game player can improve his skill purely by watching video records of an expert, without knowing what actions have been taken [12].

Among popular LfD and LfO approaches, distribution matching has served as a principled solution [2, 3, 9, 10, 13], which works by interactively estimating and minimizing the discrepancy between two

stationary distributions: one generated by the expert, and the other generated by the learning agent. To correctly estimate the distribution discrepancy, traditional approaches require *on-policy* interactions with the environment whenever the agent policy gets updated. This inefficient sampling strategy impedes wide applications of IL to scenarios where accessing transitions are expensive [14, 15]. The same challenge is aggravated in LfO, as more explorations by the agent are needed to cope with the lack of action guidance.

Towards sample-efficiency, some off-policy IL solutions have been proposed to leverage transitions cached in a replay buffer. Mostly designed for LfD, these methods either lack theoretical guarantee by ignoring a potential distribution drift [4, 16, 17], or hinge on the knowledge of expert actions to enable off-policy distribution matching [3], which makes their approach inapplicable to LfO.

To address the aforementioned limitations, in this work, we propose a LfO approach that improves sample-efficiency in a principled manner. Specifically, we derive an upper-bound of the LfO objective which dispenses with the need of knowing expert actions and can be fully optimized with *off-policy* learning. To further accelerate the learning procedure, we combine our objective with a regularization term, which is validated to pursue distribution matching between the expert and the agent from a *mode-covering* perspective. Under a mild assumption of a deterministic environment, we show that the regularization can be enforced by learning an inverse action model. We call our approach *OPOLO* (*O*ff *PO*licy *L*earning from *O*bservations). Extensive experiments on popular benchmarks show that *OPOLO* achieves state-of-the-art in terms of both asymptotic performance and sample-efficiency.

## 2   Background

We consider learning an agent in an environment of Markov Decision Process (MDP) [18], which can be defined as a tuple: $\mathcal{M} = (\mathcal{S}, \mathcal{A}, P, r, \gamma, p_0)$. Particularly, $\mathcal{S}$ and $\mathcal{A}$ are the state and action spaces; $P$ is the state transition probability, with $P(s'|s, a)$ indicating the probability of transitioning from $s$ to $s'$ upon action $a$; $r$ is the reward function, with $r(s, a)$ the immediate reward for taking action $a$ on state $s$; Without ambiguity, we consider an MDP with *infinite* horizons, with $0 < \gamma < 1$ as a discounted factor; $p_0$ is the initial state distribution. An agent follows its policy $\pi : \mathcal{S} \to \mathcal{A}$ to interact with this MDP with an objective of maximizing its expected return:

$$\max J_{\text{RL}}(\pi) := \mathbb{E}_{s_0 \sim p_0, a_i \sim \pi(\cdot|s_i), s_{i+1} \sim P(\cdot|s_i, a_i), \forall 0 \le i \le t} \Big[ \sum_{t=0}^{\infty} \gamma^t r(s_t, a_t) \Big] = \mathbb{E}_{(s,a) \sim \mu^\pi(s,a)} \Big[ r(s, a) \Big],$$

in which $\mu^\pi(s, a)$ is the *stationary state-action distribution* induced by $\pi$, as defined in Table 1.

***Learning from demonstrations*** (LfD) is a problem setting in which an agent is provided with a fixed dataset of expert demonstrations as guidance, without accessing the environment rewards . The demonstrations $\mathcal{R}_E$ contain sequences of both *states* and *actions* generated by an expert policy $\pi_E$: $\mathcal{R}_E = \{(s_0, a_0), (s_1, a_1), \cdots | a_i \sim \pi_E(\cdot|s_i), s_{i+1} \sim P(\cdot|s_i, a_i)\}$. Without ambiguity, we assume that the expert and agent are from the same MDP.

Among LfD approaches, distribution matching has been a popular choice, which minimizes the discrepancy between two stationary *state-action* distributions: one is $\mu^E(s, a)$ induced by the expert, and the other is $\mu^\pi(s, a)$ induced by the agent. Without loss of generality, we consider KL-divergence as the discrepancy measure for distribution matching, although any $f$-divergences can serve as a legitimate choice [2, 19, 20] :

$$\min J_{\text{LfD}}(\pi) := \mathbb{D}_{\textbf{KL}}[\mu^\pi(s, a) || \mu^E(s, a)]. \tag{1}$$

***Learning from observations*** (LfO) is a more challenging scenario where expert guidance $\mathcal{R}_E$ contains only *states*. Accordingly, applying distribution matching to solve LfO yields a different objective that involves *state-transition* distributions [10, 21, 9]:

$$\min J_{\text{LfO}}(\pi) := \mathbb{D}_{\textbf{KL}}[\mu^\pi(s, s') || \mu^E(s, s')]. \tag{2}$$

There exists a close connection between LfO and LfD objectives. In particular, the discrepancy between two objectives can be derived precisely as follows (see Sec 9.2 in the appendix) [10]:

$$\mathbb{D}_{\textbf{KL}}[\mu^\pi(a|s, s') || \mu^E(a|s, s')] = \mathbb{D}_{\textbf{KL}}[\mu^\pi(s, a) || \mu^E(s, a)] - \mathbb{D}_{\textbf{KL}}[\mu^\pi(s, s') || \mu^E(s, s')]. \tag{3}$$

**Remark 1.** *In a non-injective MDP, the discrepancy of* $\mathbb{D}_{\textbf{KL}}[\mu^\pi(a|s, s') || \mu^E(a|s, s')]$ *cannot be optimized without knowing expert actions. In a deterministic and injective MDP, it satisfies that* $\forall \pi : \mathcal{S} \to \mathcal{A}, \ \mathbb{D}_{\textbf{KL}}[\mu^\pi(a|s, s') || \mu^E(a|s, s')] = 0.$

| | State Distribution | State-Action Distribution | Joint Distribution | Transition Distribution | Inverse-Action Distribution |
|---|---|---|---|---|---|
| Notation | $\mu^\pi(s)$ | $\mu^\pi(s,a)$ | $\mu^\pi(s,a,s')$ | $\mu^\pi(s,s')$ | $\mu^\pi(a|s,s')$ |
| Support | $\mathcal{S}$ | $\mathcal{S} \times \mathcal{A}$ | $\mathcal{S} \times \mathcal{A} \times \mathcal{S}$ | $\mathcal{S} \times \mathcal{S}$ | $\mathcal{A} \times \mathcal{S} \times \mathcal{S}$ |
| Definition | $(1-\gamma)\sum_{t=1}^{\infty} \gamma^t \mu_t^\pi(s)$ | $\mu^\pi(s)\pi(a|s)$ | $\mu^\pi(s,a)P(s'|s,a)$ | $\int_{\mathcal{A}} \mu^\pi(s,a,s')da$ | $\frac{\mu^\pi(s,a)P(s'|s,a)}{\mu^\pi(s,s')}$ |

Table 1: Summarization on different stationary distributions, with $\mu_t^\pi(s) = p(s_t = s | s_0 \sim p_0(\cdot), a_i \sim \pi(\cdot|s_i), s_{i+1} \sim P(\cdot|s_i, a_i)), \ \forall i < t)$.

Despite the potential gap between these two objectives, the LfO objective in Eq (2) is still intuitive and valid, as it emphasizes on recovering the expert's influence on the environment by encouraging the agent to yield the desired *state-transitions*, regardless of the immediate behavior that leads to those transitions. In this work, we follow this rationale and consider Eq (2) as our learning objective, which has also been widely adopted by prior art [9, 22, 23, 24]. We will show later that pursuing this objective is sufficient to recover expertise for various challenging tasks.

A common limitation of existing LfO and LfD approaches relies in their inefficient optimization. Work along this line usually adopts a GAN-style strategy [25] to perform distribution matching. Take the representative work of *GAIL* [2] as an example, in which a discriminator $x : \mathcal{S} \times \mathcal{A} \to \mathbb{R}$ and a generator $\pi : \mathcal{S} \to \mathcal{A}$ are jointly learned to optimize a dual form of the original LfD objective:

$$\min_\pi \max_x J_{\text{GAIL}}(\pi, x) := \mathbb{E}_{\mu^E(s,a)}[\log(x(s,a))] + \mathbb{E}_{\mu^\pi(s,a)}[\log(1 - x(s,a))].$$

During optimization, *on-policy* transitions in the MDP are used to estimate expectations over $\mu^\pi$. It requires new environment interactions whenever $\pi$ gets updated and is thus sample inefficient. This inconvenience is echoed in the work of LfO, which inherits the same spirit of on-policy learning [10, 9]. In pursuit of sample-efficiency, some off-policy solutions have been proposed. These methods, however, either lack theoretical guarantee [17, 4], or rely on the expert actions [4, 3], which makes them inapplicable to LfO. We will provide more explanations in Sec 9.8 in the appendix.

To improve the sample-efficiency of LfO with a principled solution, in the next section we show how we explicitly introduce an off-policy distribution into the LfO objective, from which we derive a feasible upper-bound that enables *off-policy* optimization without the need of accessing expert actions.

## 3 *OPOLO*: Off-Policy Learning from Observations

### 3.1 Surrogate Objective
The idea of re-using cached transitions to improve sample-efficiency has been adopted by many RL algorithms [7, 26, 27, 28]. In the same spirit, we start by introducing an *off-policy* distribution $\mu^R(s,a)$, which is induced by a dataset $\mathcal{R}$ of historical transitions. Choosing KL-divergence as a discrepancy measure, we obtain an upper-bound of the LfO objective by involving $\mu^R(s,a)$ (see Sec 9.1 in the appendix for proof):

$$\mathbb{D}_{\mathbf{KL}}\left[\mu^\pi(s,s')||\mu^E(s,s')\right] \leq \mathbb{E}_{\mu^\pi(s,s')}\left[\log \frac{\mu^R(s,s')}{\mu^E(s,s')}\right] + \mathbb{D}_{\mathbf{KL}}\left[\mu^\pi(s,a)||\mu^R(s,a)\right]. \quad (4)$$

As a result, the LfO objective can be optimized by minimizing the RHS of Eq (4). Although widely adopted for its interpretability, KL divergence can be tricky to estimate due to issues of biased gradients [29, 3]. To avoid the potential difficulty in optimization, we further substitute the term $\mathbb{D}_{\mathbf{KL}}[\mu^\pi(s,a)||\mu^R(s,a)]$ in Eq (4) by a more aggressive $f$-divergence, with $f(x) = \frac{1}{2}x^2$, which serves as an upper-bound of the KL-divergence (See Sec 9.4 in the appendix):

$$\mathbb{D}_{\mathbf{KL}}[P||Q] \leq \mathbb{D}_f[P||Q]. \quad (5)$$

Our choice of $f$-divergence can be considered as a variant of Pearson $\chi^2$-divergence with a constant shift, which has also been adopted as a valid measure of distribution discrepancies [30, 31]. Compared with KL-divergence, this $f$-divergence enables unbiased estimation without deteriorating the optimality, whose advantages will become increasingly visible in Section 3.2.

Built upon the above transformations, we reach an objective that serves as an effective upper-bound of $\mathbb{D}_{\mathbf{KL}}[\mu^\pi(s,s')||\mu^E(s,s')]$:

$$\min_{\pi} J_{\text{opolo}}(\pi) := \mathbb{E}_{\mu^{\pi}(s,s')} \left[ \log \frac{\mu^R(s,s')}{\mu^E(s,s')} \right] + \mathbb{D}_f[\mu^{\pi}(s,a) || \mu^R(s,a)]. \tag{6}$$

## 3.2 Off-Policy Transformation

Optimization Eq (6) is still *on-policy* and induces additional challenges through the term $\mathbb{D}_f[\mu^{\pi}(s,a) || \mu^R(s,a)]$. However, we show that it can be readily transformed into off-policy learning. We first leverage the dual-form of an $f$-divergence [32]:

$$-\mathbb{D}_f[\mu^{\pi}(s,a) || \mu^R(s,a)] = \inf_{x:S \times A \to R} \mathbb{E}_{(s,a) \sim \mu^{\pi}}[-x(s,a)] + \mathbb{E}_{(s,a) \sim \mu^R}[f_*(x(s,a))],$$

and use this dual transformation to rewrite Eq (6):

$$\min_{\pi} J_{\text{opolo}}(\pi) \equiv \max_{\pi} \ \mathbb{E}_{\mu^{\pi}(s,s')} \left[ -\log \frac{\mu^R(s,s')}{\mu^E(s,s')} \right] - \mathbb{D}_f[\mu^{\pi}(s,a) || \mu^R(s,a)]$$

$$\equiv \max_{\pi} \min_{x:S \times A \to R} J_{\text{opolo}}(\pi, x) := \mathbb{E}_{\mu^{\pi}(s,a,s')} \left[ \log \frac{\mu^E(s,s')}{\mu^R(s,s')} - x(s,a) \right] + \mathbb{E}_{\mu^R(s,a)}[f_*(x(s,a))]. \tag{7}$$

If we consider a synthetic reward as $r(s,a,s') = \log \frac{\mu^E(s,s')}{\mu^R(s,s')} - x(s,a)$, the first term in Eq (7) resembles an RL return function: $\hat{J}(\pi) = \mathbb{E}_{(s,a,s') \sim \mu^{\pi}(s,a,s')}[r(s,a,s')]$. Observing this similarity, we turn to learning a $Q$-function by applying a change of variables:

$$Q(s,a) = \mathbb{E}_{s' \sim P(\cdot|s,a),a' \sim \pi(\cdot|s')} \left[ -x(s,a) + \log \frac{\mu^E(s,s')}{\mu^R(s,s')} + \gamma Q(s',a') \right].$$

Equivalently, this $Q$ function is a fixed point of a variant Bellman operator $\mathcal{B}^{\pi}Q$:

$$Q(s,a) = -x(s,a) + \mathbb{E}_{s' \sim P(\cdot|s,a),a' \sim \pi(\cdot|s')} \left[ \log \frac{\mu^E(s,s')}{\mu^R(s,s')} + \gamma Q(s',a') \right] = -x(s,a) + \mathcal{B}^{\pi}Q(s,a).$$

Rewriting $x(s,a) = (\mathcal{B}^{\pi}Q - Q)(s,a)$ and applying it back to Eq (7), we finally remove the *on-policy* expectation by a series of telescoping (see Sec 9.6 in the appendix for derivation):

$$\max_{\pi} \min_{x:S \times A \to R} J_{\text{opolo}}(\pi, x) \equiv \max_{\pi} \min_{Q:S \times A \to R} J_{\text{opolo}}(\pi, Q)$$

$$:= \mathbb{E}_{(s,a,s') \sim \mu^{\pi}(s,a,s')}[\log \frac{\mu^E(s,s')}{\mu^R(s,s')} - (\mathcal{B}^{\pi}Q - Q)(s,a)] + \mathbb{E}_{(s,a) \sim \mu^R(s,a)}[f_*((\mathcal{B}^{\pi}Q - Q)(s,a))]$$

$$= (1 - \gamma)\mathbb{E}_{s_0 \sim p_0, a_0 \sim \pi(\cdot|s_0)}[Q(s_0,a_0)] + \mathbb{E}_{(s,a) \sim \mu^R(s,a)}[f_*((\mathcal{B}^{\pi}Q - Q)(s,a))]. \tag{8}$$

A similar rationale has also been the key component of *distribution error correction* (DICE) [30, 31, 33]. Based on the above transformation, we propose our main objective:

$$\max_{\pi} \min_{Q:S \times A \to R} J_{\text{opolo}}(\pi, Q) := (1 - \gamma)\mathbb{E}_{s_0 \sim p_0, a_0 \sim \pi(\cdot|s_0)}[Q(s_0,a_0)] + \mathbb{E}_{\mu^R(s,a)}[f_*((\mathcal{B}^{\pi}Q - Q)(s,a))]. \tag{9}$$

Specifically, when $f(x) = f^*(x) = \frac{1}{2}x^2$, the second term $\mathbb{E}_{\mu^R(s,a)}[f_*((\mathcal{B}^{\pi}Q - Q)(s,a))]$ is reminiscent of an Bellman error, for which we can have unbiased estimation by mini-batch gradients.

Given access to the *off-policy* distribution $\mu^R(s,a)$ and the initial distribution $p_0$, optimization (9) can be efficiently realized once we resolve the term $\log \frac{\mu^E(s,s')}{\mu^R(s,s')}$ contained in $\mathcal{B}^{\pi}Q(s,a)$.

## 3.3 Adversarial Training with Off-Policy Experience

We can take the advantage of GAN training [25] to estimate the term $\log \frac{\mu^E(s,s')}{\mu^R(s,s')}$ inside $\mathcal{B}^{\pi}Q(s,a)$, by learning a discriminator $D$:

$$\max_{D:S \times S \to \mathbb{R}} \mathbb{E}_{(s,s') \sim \mu^E(s,s')} \left[ \log(D(s,s')) \right] + \mathbb{E}_{(s,s') \sim \mu^R(s,s')} \left[ \log(1 - D(s,s')) \right],$$

which upon training to optimality, satisfies $\log(\frac{\mu_E(s,s')}{\mu^R(s,s')}) = \log D^*(s,s') - \log(1 - D^*(s,s'))$.

Unlike prior art [2, 9, 4] that requires estimating the ratio of $\log \frac{\mu^E}{\mu^{\pi}}$, the discriminator in our case is designed to be off-policy in accordance with our proposed objective. Up to this step, optimization (9) can be achieved by interactively optimizing $Q$, $\pi$, and $D$ with pure off-policy learning.

### 3.4 Policy Regularization as Forward Distribution Matching

Optimization 9 essentially minimizes an upper-bound of the *inverse* KL divergence $\mathbb{D}_{\mathbf{KL}}[\mu^\pi(s, s')||\mu^E(s, s')]$, which is known to encourage a *mode-seeking* behavior [34]. Although mode-seeking is more robust to covariate-drift than *mode-covering* (such as behavior cloning), it requires sufficient explorations to find a reasonable state-distribution, especially at early learning stages. On the other hand, a mode-covering strategy has merits in quickly minimizing discrepancies on the *expert* distribution, by optimizing a *forward* KL-divergence such as $\mathbb{D}_{\mathbf{KL}}[\pi_E(a|s)||\pi(a|s)]$.

To combine the advantages of both, in this section we show how we further speed up the learning procedure from a *mode-covering* perspective, without deteriorating the efficacy of our main objective. To achieve this goal, we first derive an optimizable lower-bound from a *mode-covering* objective:

$$\mathbb{D}_{\mathbf{KL}}[\pi_E(a|s)||\pi(a|s)] = \mathbb{D}_{\mathbf{KL}}[\mu^E(s'|s)||\mu^\pi(s'|s)] + \mathbb{D}_{\mathbf{KL}}[\mu^E(a|s, s')||\mu^\pi(a|s, s')], \quad (10)$$

in which we define $\mu^\pi(s'|s) = \int_{\mathcal{A}} \pi(a|s)P(s'|s, a)da$ as the *conditional* state transition distribution induced by $\pi$, likewise for $\mu^E(s'|s)$ (see Sec 9.5 in the appendix).

Similar to Remark 1, the discrepancy $\mathbb{D}_{\mathbf{KL}}[\mu^E(a|s, s')||\mu^\pi(a|s, s')]$ is not optimizable without knowing expert actions. However, under some mild assumptions, we found it feasible to optimize the other term $\mathbb{D}_{\mathbf{KL}}[\mu^E(s'|s)||\mu^\pi(s'|s)]$ by enforcing a policy regularization:

**Remark 2.** *In a deterministic MDP, assuming the support of $\mu^E(s, s')$ is covered by $\mu^R(s, s')$, s.t. $\mu^E(s, s') > 0 \implies \mu^R(s, s') > 0$, then regulating policy using $\mu^R(\cdot|s, s')$ minimizes $\mathbb{D}_{\mathbf{KL}}[\mu^E(s'|s)||\mu^\pi(s'|s)]$ (See Sec 9.5.2 in supplementary for a detailed discussion):*

$$\exists \tilde{\pi} : \mathcal{S} \to \mathcal{A}, \ s.t. \ \forall (s, s') \sim \mu^E(s, s'), \ \tilde{\pi}(\cdot|s) \propto \mu^R(\cdot|s, s') \implies \tilde{\pi} = \arg\min_\pi \mathbb{D}_{\mathbf{KL}}[\mu^E(s'|s)||\mu^\pi(s'|s)].$$

Intuitively, when expert labels are unavailable, this regularization can be considered as performing *states matching*, by encouraging the policy to yield actions that lead to desired footprints. Given a transition $s \to s'$ from the *expert observations*, a conditional distribution $\mu^R(\cdot|s, s')$ only has *support* on actions that yield this transition $s \to s'$. Therefore, following this regularization avoids the policy from drifting to undesired states.

In practice, we can estimate $\mu^R(\cdot|s, s')$ by learning an inverse action model $P_I$ using off-policy transitions from $\mu^R(s, a, s')$ to optimize the following (See Sec 9.5.3 in the appendix):

$$\max_{P_I: \mathcal{S} \times \mathcal{S} \to \mathcal{A}} -\mathbb{D}_{\mathbf{KL}}[\mu^R(a|s, s')||P_I(a|s, s')] \equiv \max_{P_I: \mathcal{S} \times \mathcal{S} \to \mathcal{A}} \mathbb{E}_{(s, a, s') \sim \mu^R(s, a, s')}[\log P_I(a|s, s')]. \quad (11)$$

### 3.5 Algorithm

Based on all the abovementioned building blocks, we now introduce *OPOLO* in Algorithm 1. *OPOLO* involves learning a policy $\pi$, a critic $Q$, a discriminator $D$, and an inverse action regularizer $P_I$, all of which can be done through off-policy training.

In particular, $\pi$ and $Q$ is jointly learned to find a saddle-point solution to optimization (9). The discriminator $D$ assists this process by estimating a density ratio $\log \frac{\mu^E(s, s')}{\mu^R(s, s')}$. For better empirical performance, we adopt $-\log(1 - D(s, s'))$ as the discriminator's output, which corresponds to a constant shift inside the logarithm term, in that $\log(\frac{\mu^E(s, s')}{\mu^R(s, s')} + 1) = -\log(1 - D^*(s, s'))$. The inverse action model $P_I$ serves as a regularizer to infer proper actions on the *expert observation distribution* to encourage *mode-covering*. We defer more implementation details to Sec 9.7 in the appendix.

## 4 Related Work

Recent development on imitation learning can be divided into two categories:

***Learning from Demonstrations*** (LfD) traces back to behavior cloning (***BC***) [35], in which a policy is pre-trained to minimize the prediction error on expert demonstrations. This approach is inherent with issues such as distribution shift and regret propagations. To address these limitations, [1] proposed a no-regret IL approach called *DAgger*, which however requires online access to oracle corrections. More recent LfD approaches favor Inverse reinforcement learning (***IRL***) [8], which work by seeking a reward function that guarantees the superiority of expert demonstrations, based on which regular RL algorithms can be used to learn a policy [36, 37]. A representative instantiation of IRL is Generative Adversarial Imitation Learning (***GAIL***) [2]. It defines IL as a distribution matching problem and leverages the GAN technique [25] to minimize the Jensen-Shannon divergence between distributions induced by the expert and the learning policy. The success of *GAIL* has inspired many other related

---

**Algorithm 1** *O*ff-*PO*licy *L*earning from *O*bservations (*OPOLO*)

---
**Input:** expert observations $\mathcal{R}_E$, off-policy-transitions $\mathcal{R}$, initial states $\mathcal{S}_0$, $f$- function,
       policy $\pi_\theta$, critic $Q_\phi$, discriminator $D_w$, inverse action model $P_{I_\varphi}$, learning rate $\alpha$.
**for** $n = 1, \ldots$ **do**
    sample trajectory $\tau \sim \pi_\theta$, $\mathcal{R} \leftarrow \mathcal{R} \cup \tau$
    update $D_w$:    $w \leftarrow w + \alpha \hat{\mathbb{E}}_{(s,s')\sim \mathcal{R}_E}[\nabla_w \log(D_w(s,s'))] + \hat{\mathbb{E}}_{(s,s')\sim \mathcal{R}}[\nabla_w \log(1 - D_w(s,s'))]$.
                 set $r(s,s') = -\log(1 - D_w(s,s'))$.
    update $P_{I_\varphi}$:      $\varphi \leftarrow \varphi + \alpha \hat{\mathbb{E}}_{(s,a,s')\sim \mathcal{R}}[\nabla_\varphi \log(P_{I_\varphi}(a|s,s'))]$.
    update $\pi_\theta$ and $Q_\phi$ :
$$J(\pi_\theta, Q_\phi) = (1-\gamma)\hat{\mathbb{E}}_{s\sim\mathcal{S}_0}[Q_\phi(s,\pi_\theta(s))] + \hat{\mathbb{E}}_{(s,a,s')\sim\mathcal{R}}\Big[f^*\Big(r(s,s') + \gamma Q_\phi(s',\pi_\theta(s')) - Q_\phi(s,a)\Big)\Big].$$

$$J_{\text{Reg}}(\pi_\theta) = \mathbb{E}_{(s,s')\sim\mathcal{R}_E, a\sim P_{I_\varphi}(\cdot|s,s')}[\log \pi_\theta(a|s)].$$
              $\phi \leftarrow \phi - \alpha J_{\nabla\phi}(\pi_\theta, Q_\phi); \quad \theta \leftarrow \theta + \alpha\big(J_{\nabla\theta}(\pi_\theta, Q_\phi) + J_{\nabla\theta}J_{\text{Reg}}(\pi_\theta)\big).$
**end for**

---

work, including adopting different RL frameworks [4], or choosing different divergence measures [13, 5, 38] to enhance the effectiveness of imitation learning. Most work along this line focuses on *on-policy* learning, which is a sample-costly strategy.

As an *off-policy* extension of *GAIL* , *DAC* [4] improves the sample-efficiency by re-using previous samples stored in a relay buffer rather than on-policy transitions. Similar ideas of reusing cached transitions can be found in [16]. One limitation of these approaches is that they neglected the discrepancy induced when replacing the on-policy distribution with off-policy approximations, which results in a deviation from their proposed objective. Another off-policy imitation learning approach is ValueDICE [3], which inherits the idea of DICE [30] to transform an on-policy LfD objective to an off-policy one. This approach, however, requires the information of expert actions, which otherwise makes off-policy estimation unreachable in a model-free setting. Therefore, their approach is not directly applicable to LfO. We have analyzed this dilemma in Sec 9.8 in the appendix.

***Learning from Observations*** (LfO) tackles a more challenging scenario where expert actions are unavailable. Work alone this line falls into *model-free* and *model-based* approaches. *GAIfO* [9] is a model-free solution which applies the principle of *GAIL* to learn a discriminator with state-only inputs. *IDDM* [10] further analyzed the theoretical gap between the LfD and LfO objectives, and proved that a lower-bound of this gap can be somewhat alleviated by maximizing the mutual-information between $(s, (a, s'))$, given an on-policy distribution $\mu^\pi(s, a, s')$. Its performance is comparable to *GAIL*. [24] assumed that the given observation sequences are ranked by superiority, based on which a reward function is designed for policy learning. Similar to *GAIL*, the sample efficiency of these approaches is suboptimal due to their *on-policy* strategy.

Model-based LfO can be further organized into learning a *forward* [23, 39] dynamics model or an *inverse* action model [17, 21]. Especially, [23] proposed a forward model solution to learn time-dependent policies for finite-horizon tasks, in which the number of policies to be learned equals the number of transition steps. This approach may not be suitable for tasks with long or infinite horizons. Behavior cloning from observations (*BCO*) [17] learns an inverse model to infer actions missing from the expert dataset, after which behavior cloning is applied to learn a policy. Besides the common issues faced by BC, this strategy does not guarantee that the ground-truth expert actions can be recovered, unless is a deterministic and injective MDP is assumed. Some other recent work focused on different problem settings than ours, in which the expert observations are collected with different transition dynamics [40] or from different viewpoints [21, 41, 42]. Readers are referred to [11] for further discussions of LfO.

## 5 Experiments

We compare *OPOLO* against state-of-the-art LfD and LfO approaches on MuJuCo benchmarks, which are locomotion tasks in continuous state-action space. In accordance with our assumption in Sec 3.4, these tasks have *deterministic* dynamics. Original rewards are removed from all benchmarks to fit into an IL scenario. For each task, we collect 4 trajectories from a pre-trained expert policy. All illustrated results are evaluated across 5 random seeds.

**Baselines**: We compared *SAIL* against 7 baselines. We first selected 5 representative approaches from prior work: *GAIL* (on-policy LfD), *DAC* (off-policy LfD), *ValueDICE* (off-policy LfD), *GAIfO* (on-policy LfO), and *BCO* (off-policy LfO). We further designed two strong *off-policy* approaches, Specifically, we built *DACfO*, which is a variation of *DAC* that learns the discriminator on $(s, s')$ instead of $(s, a)$, and *ValueDICEfO*, which is built based on *ValueDICE*. Instead of using ground-truth expert actions, *ValueDICEfO* learns an inverse model by optimizing Eq (11), and uses the approximated actions generated by the inverse model to fit an LfO problem setting. To the best of our knowledge, *DACfO* and *ValueDICEfO* have not been investigated by any prior art. Among these baselines, *GAIL*, *DAC*, and *ValueDICE* are provided with both expert *states* and *actions*, while all other approaches only have access to expert *states*. More experimental details can be found in the supplementary material.

Our experiments focus on answering the following important questions:

1. *Asymptotic performance*: Is *OPOLO* able to achieve expert-level performance given a limited number of expert observations?
2. *Sample efficiency*: Can *OPOLO* recover expert policy using less interactions with the environment, compared with the state-of-the-art?
3. *Effects of the inverse action regularization*: Does the inverse action regularization useful in speeding up the imitation learning process?
4. *Sensitivity of the choice of $f$-divergence*: Can *OPOLO* perform well given different $f$ functions?

## 5.1  Performance Comparison

*OPOLO* can recover expert performance given a fixed budget of expert observations. As shown in Figure 1, *OPOLO* reaches (near) optimal performance in all benchmarks. For simpler tasks such as *Swimmer* and *InvertedPendulum*, most baselines can successfully recover expertise. For other complex tasks with high state-action space, on-policy baselines, such as *GAIL* and *GAIfO*, are struggling to reach their asymptotic performance within a limited number of interactions, As shown in Figure 2, the off-policy baseline *BCO* is prone to sub-optimality due to its behavior cloning-like strategy, On the other hand, the performance of *ValueDICEfO* can be deteriorated by potential action-drifts, as the inferred actions are not guaranteed to recover expertise. For fair comparison, performance of all *off-policy* approaches are summarized in Table 2 given a fixed number of interaction steps.

The asymptotic performance of *OPOLO* is 1) superior to *DACfO* and *ValueDICEfO*, 2) comparable to *DAC*, and 3) is more robust against overfitting compared with *ValueDICE*, whereas both *DAC* and *ValueDICE* enjoy the advantage of off-policy learning and extra action guidance.

| Env | HalfCheetah | Hopper | Walker | Swimmer | Ant |
|---|---|---|---|---|---|
| *BCO* | 3881.10±938.81 | 1845.66±628.41 | 421.24±135.18 | 256.88±4.52 | 1529.54±980.86 |
| *OPOLO-x* | **7632.80±128.88** | 3581.85±19.08 | 3947.72±97.88 | 246.62±1.56 | 5112.04±321.42 |
| *OPOLO* | 7336.96±117.89 | 3517.39±25.16 | 3803.00±979.85 | 257.38±4.28 | **5783.57±651.98** |
| *DAC* | 6900.00±131.24 | 3534.42±10.27 | **4131.05±174.13** | 232.12±2.04 | 5424.28±594.82 |
| *DACfO* | 7035.63±444.14 | 3522.95±93.15 | 3033.02±207.63 | 185.28±2.67 | 4920.76±872.66 |
| *ValueDICE* | 5696.94±2116.94 | **3591.37±8.60** | 1641.58±1230.73 | 262.73±7.76 | 3486.87±1232.25 |
| *ValueDICEfO* | 4770.37±644.49 | 3579.51±10.23 | 431.00±140.87 | **265.05±3.45** | 75.08±400.87 |
| Expert | 7561.78±181.41 | 3589.88±2.43 | 3752.67±192.80 | 259.52±1.92 | 5544.65±76.11 |
| $(\mathcal{S}, \mathcal{A})$ | $(17, 6)$ | $(11, 3)$ | $(17, 6))$ | $(8, 2)$ | $(111, 8)$ |

Table 2: Evaluated performance of *off-policy* approaches. Results are averaged over 50 trajectories.

## 5.2  Sample Efficiency

*OPOLO* is comparable with and sometimes superior to *DAC* in all evaluated tasks, and is much more sample-efficient than *on-policy* baselines. As shown in Figure 1, the sample-efficiency of *OPOLO* is emphasized by benchmarks with high state-action dimensions. In particular, for tasks such as *Ant* or *HalfCheetah*, the performance curves of *on-policy* baselines are barely improved at early learning stages. One intuition is that they need more explorations to build the current support of the learning policy, which cannot benefit from cached transitions. For these challenging tasks, *OPOLO* is even more sample-efficient than *DAC* that has the guidance of expert actions. We ascribe this improvement to the *mode-covering* regularization of *OPOLO* enforced by its inverse action model, whose effect will be further analyzed in Sec 5.3. Meanwhile, other off-policy approaches such as *BCO* and

*ValueDICEfO*, are prone to overfitting and performance degradation (as shown in Figure 2), which indicates that the effect of the inverse model alone is not sufficient to recover expertise. On the other hand, the *ValueDICE* algorithm, although being sample-efficient, is not designed to address LfO and requires expert actions.

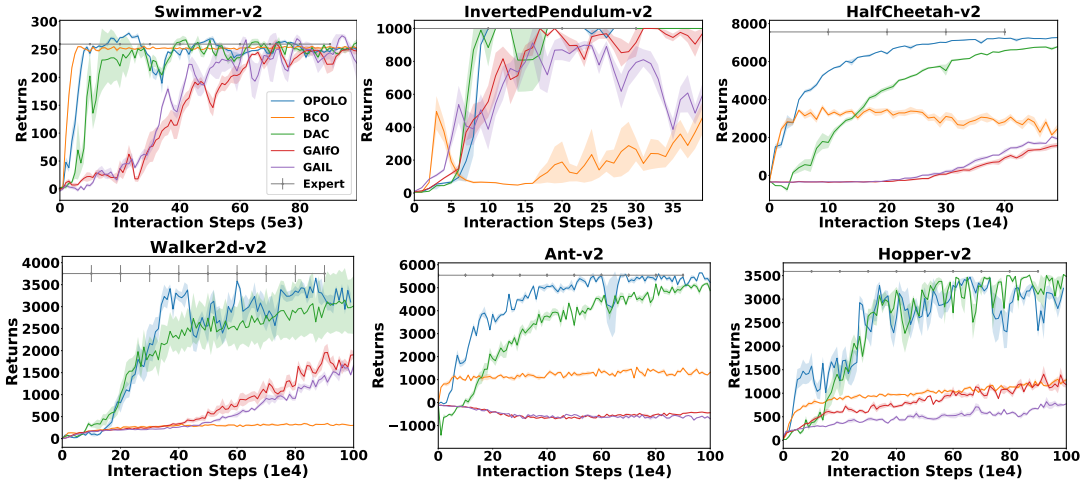

Figure 1: Interaction steps ($x$-axis) versus learning performance ($y$-axis). Compared with *GAIL*, *BCO*, *GAIfO*, and *DAC*, our proposed approach (*OPOLO*) is the most sample-efficient to reach expert-level performance (Grey horizontal line).

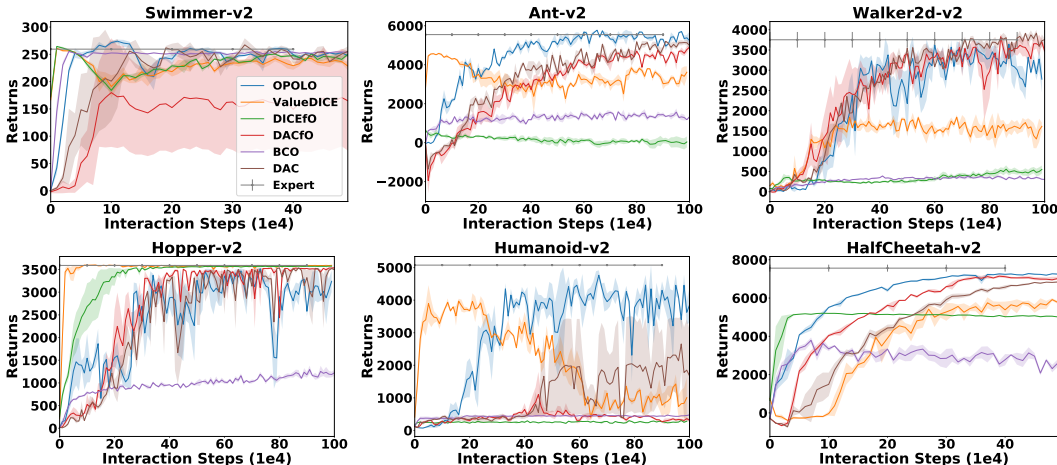

Figure 2: Compared with strong off-policy baselines, *OPOLO* is the only approach that consistently achieves competitive performance regarding both sample-efficiency and asymptotic performance across all tasks, without accessing expert actions.

## 5.3 Ablation Study

In this section, we further analyze the effects of the inverse action regularization by a group of ablation studies. Especially, we implement a variant of *OPOLO* that does not learn an inverse action model to regulate the policy update. We compare this approach, dubbed as *OPOLO-x*, against our original approach as well as the *DAC* algorithm.

***Effects on Sample efficiency:*** Performance curves in Figure 3 show that removing the inverse action regularization from *OPOLO* slightly affects its sample-efficiency, although the degraded version is still comparable to *DAC*. This impact is more visible in challenging tasks such as *HalfCheetah* and *Ant*. From another perspective, the same phenomenon indicates that an inverse action regularization is beneficial for accelerating the IL process, especially for games with high observation-space. An intuitive exploration is that, while our main objective serves as a driving force for *mode-seeking*, a regularization term assists by encouraging the policy to perform *mode-covering*. Combing these two motivations leads to a more efficient learning strategy.

***Effects on Performance:*** Given a reasonable number of transition steps, the effects of an inverse-action model are less obvious regarding the asymptotic performance. As shown in Table 2, *OPOLO-x* is mostly comparable to *OPOLO* and *DAC*. This implies that the effect of the *state-covering* regularization will gradually fade out once the policy learns a reasonable state distribution. From another perspective, it indicates that following our main objective alone is sufficient to recover expert-level performance. Comparing with *BCO* which uses the inverse model solely for behavior cloning, we find it more effective when serving as a regularization to assist distribution matching from a *forward* direction.

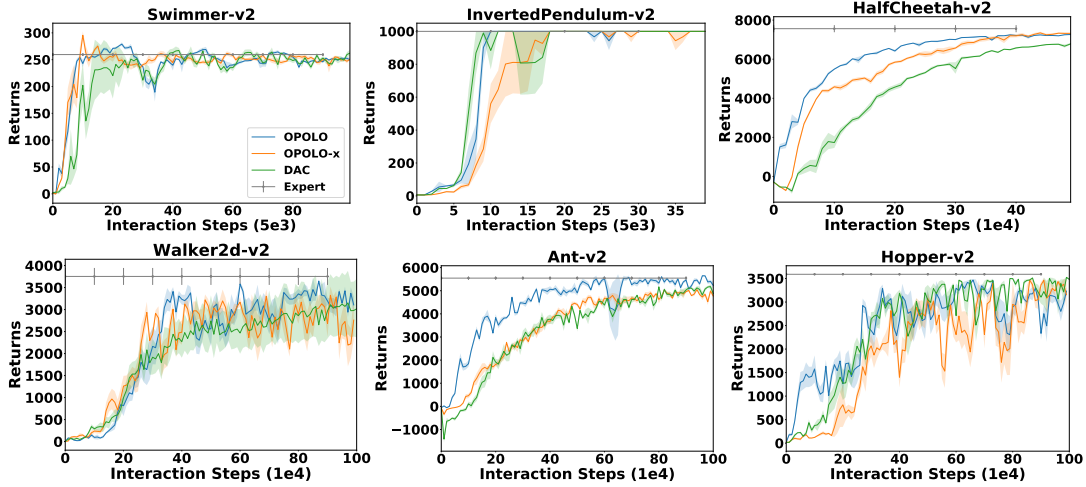

Figure 3: Removing the inverse action regularization (*OPOLO-x*) results in slight efficiency drop, although its performance is still comparable to *OPOLO* and *DAC*.

## 5.4 Sensitivity Analysis

To analyze the effects of different $f$-functions on the performance of the proposed approach, we explored a family of $f$-divergence where $f(x) = \frac{1}{p}|x|^p$, $f^*(y) = \frac{1}{q}|y|^q$, s.t. $\frac{1}{p} + \frac{1}{q} = 1, p, q > 1$, as adopted by *DualDICE* [30]. Evaluation results show that *OPOLO* yields reasonable performance across different $f$-functions, although our choice ($q = p = 2$) turns out to be most stable. Results using the *Ant* task is illustrated in Figure 4.

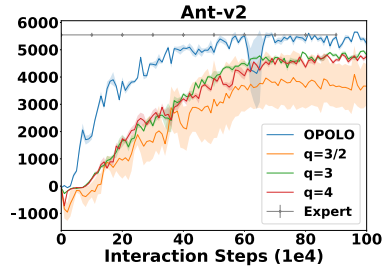

Figure 4: Performance of *SAIL* given different $f$-functions.

## 6 Conclusions

Towards sample-efficient imitation learning from observations (LfO), we proposed a *principled* approach that performs imitation learning by accessing only a limited number of expert observations. We derived an upper-bound of the original LfO objective to enable efficient off-policy optimization, and augment the objective with an inverse action model regularization to speeds up the learning procedure. Extensive empirical studies are done to validate the proposed approach.

## 7 Acknowledgments

This research was jointly supported by the National Science Foundation IIS-1749940, and the Office of Naval Research N00014-20-1-2382. We would like to thank Dr. Boyang Liu and Dr. Junyuan Hong (Michigan State University) for providing insightful comments. We also appreciate Dr. Mengying Sun (Michigan State University) for her assistance in proofreading the manuscript.

## 8 Broader Impact

The success of Imitation Learning (IL) is crucial for realizing robotic intelligence. Serving as an effective solution to a practical IL setting, *OPOLO* has a promising future in various applications, including robotics control [43], game-playing [6], autonomous driving [14], algorithmic trading [44], to name just a few.

On one hand, *OPOLO* provides an working evidence of sample-efficient IL. *OPOLO* costs less environment interactions compared with conventional IL approaches. For tasks where taking real actions can be expensive (high-frequency trading) or dangerous (autonomous driving), using less interactions for imitation learning is a crucial requirement for successful applications.

On the other hand, *OPOLO* validates the feasibility of learning from incomplete guidance, and can enable IL in applications where expert demonstrations are costly to access. Moreover, *OPOLO* is more resemblant to human intelligence, as it can recover expertise simply by learning from expert observations. In general, *OPOLO* has a strong impact on the advancement of IL, from the perspective of both theoretical and empirical studies.

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
