[Supplementary Material]

# 9 Appendix

For all the following derivations, we use $\mathbb{D}_{\mathbf{KL}}[P(X)||Q(X)]$ to denote the KL-divergence between two distributions $P$ and $Q$:

$$\mathbb{D}_{\mathbf{KL}}[P(X)||Q(X)] = \mathbb{E}_{x \sim p(x)} \log \frac{p(x)}{q(x)} = \int_X p(x) \log \frac{p(x)}{q(x)} dx.$$

Accordingly, when $P(X|Z)$ and $Q(X|Z)$ are *conditional* distributions, $\mathbb{D}_{\mathbf{KL}}[P||Q]$ denotes their *conditional* KL-divergence:

$$\mathbb{D}_{\mathbf{KL}}[P(X|Z)||Q(X|Z)] = \int_{Z \times X} p(z)p(x|z) \log \frac{p(x|z)}{q(x|z)} dxdz.$$

For simplicity, we will equivalently use $\mathbb{E}_{x \sim p(x)}[\cdot]$ and $\mathbb{E}_{p(x)}[\cdot]$ to denote certain expectation in which $x$ is sampled from the distribution $P(X)$.

## 9.1 Derivation of Surrogate Objective

We first refer Lemma 1 from [10] for a complete presentation:

**Lemma 1.**

$$\mathbb{D}_{\mathbf{KL}}[\mu^\pi(s, a, s')||\mu^E(s, a, s')] = \mathbb{D}_{\mathbf{KL}}[\mu^\pi(s, a)||\mu^E(s, a))].$$

*Proof.*

$$\begin{aligned}
\mathbb{D}_{\mathbf{KL}}[\mu^\pi(s, a, s')||\mu^E(s, a, s')] &= \int_{\mathcal{S} \times \mathcal{A} \times \mathcal{S}} \mu^\pi(s, a, s') \log \frac{\mu^\pi(s, a) \cdot P(s'|s, a)}{\mu^E(s, a) \cdot P(s'|s, a)} ds' dads \\
&= \int_{\mathcal{S} \times \mathcal{A} \times \mathcal{S}} \mu^\pi(s, a, s') \log \frac{\mu^\pi(s, a)}{\mu^E(s, a)} ds' dads \\
&= \int_{\mathcal{S} \times \mathcal{A}} \mu^\pi(s, a) \log \frac{\mu^\pi(s, a)}{\mu^E(s, a)} dads \\
&= \mathbb{D}_{\mathbf{KL}}[\mu^\pi(s, a)||\mu^E(s, a)].
\end{aligned}$$

$\square$

**Lemma 2.**

$$\mathbb{D}_{\mathbf{KL}}[\mu^\pi(s, s')||\mu^E(s, s')] \leq \mathbb{D}_{\mathbf{KL}}[\mu^\pi(s, a)||\mu^E(s, a)].$$

*Proof.* As defined in Table 1, $\mu^\pi(a|s, s')$ is the inverse-action transition probability induced by policy $\pi$:

$$\mu^\pi(a|s, s') = \frac{\mu^\pi(s, a, s')}{\mu^\pi(s, s')} = \frac{\cancel{\mu^\pi(s)} \pi(a|s) P(s'|s, a)}{\int_\mathcal{A} \cancel{\mu^\pi(s)} \pi(\bar{a}|s) P(s'|s, \bar{a}) d\bar{a}} = \frac{\pi(a|s) P(s'|s, a)}{\int_\mathcal{A} \pi(\bar{a}|s) P(s'|s, \bar{a}) d\bar{a}}.$$

Based on this notion, we can derive:

$$\begin{aligned}
&\mathbb{D}_{\mathbf{KL}}[\mu^\pi(s, a)||\mu^E(s, a)] \\
&= \underbrace{\mathbb{D}_{\mathbf{KL}}[\mu^\pi(s, a, s')||\mu^E(s, a, s')]}_{\text{Lemma 1}} \\
&= \int_{\mathcal{S} \times \mathcal{A} \times \mathcal{S}} \mu^\pi(s, a, s') \log \frac{\mu^\pi(s, a, s')}{\mu^E(s, a, s')} ds' dads \\
&= \int_{\mathcal{S} \times \mathcal{A} \times \mathcal{S}} \mu^\pi(s, s') \mu^\pi(a|s, s') \log \frac{\mu^\pi(s, s') \times \mu^\pi(a|s, s')}{\mu^E(s, s') \times \mu^E(a|s, s')} ds' dads
\end{aligned}$$

$$= \int_{\mathcal{S} \times \mathcal{A} \times \mathcal{S}} \mu^\pi(s, s') \mu^\pi(a|s, s') \log \frac{\mu^\pi(s, s')}{\mu^E(s, s')} ds' dads + \int_{\mathcal{S} \times \mathcal{A} \times \mathcal{S}} \mu^\pi(s, s') \mu^\pi(a|s, s') \log \frac{\mu^\pi(a|s, s')}{\mu^E(a|s, s')} ds' dads$$

$$= \int_{\mathcal{S} \times \mathcal{A} \times \mathcal{S}} \mu^\pi(s, s') \log \frac{\mu^\pi(s, s')}{\mu^E(s, s')} ds' ds + \mathbb{D}_{\mathbf{KL}}[\mu^\pi(a|s, s')||\mu^E(a|s, s')]$$

$$= \mathbb{D}_{\mathbf{KL}}[\mu^\pi(s, s')||\mu^E(s, s')] + \mathbb{D}_{\mathbf{KL}}[\mu^\pi(a|s, s')||\mu^E(a|s, s')] \tag{12}$$

$$\geq \mathbb{D}_{\mathbf{KL}}[\mu^\pi(s, s')||\mu^E(s, s')].$$

$\square$

Based on Lemma2, we can derive the upper-bound of our original objective:

**Theorem 1** (Surrogate Objective as the Divergence Upper-bound)**.**

$$\mathbb{D}_{\mathbf{KL}}[\mu^\pi(s, s')||\mu^E(s, s')] \leq \mathbb{E}_{\mu^\pi(s, s')}[\log \frac{\mu^R(s, s')}{\mu^E(s, s')}] + \mathbb{D}_{\mathbf{KL}}[\mu^\pi(s, a)||\mu^R(s, a)].$$

*Proof.*

$$
\begin{aligned}
\mathbb{D}_{\mathbf{KL}}[\mu^\pi(s, s')||\mu^E(s, s')] &= \int_{\mathcal{S} \times \mathcal{S}} \mu^\pi(s, s') \log \frac{\mu^\pi(s, s')}{\mu^E(s, s')} ds ds' \\
&= \int_{\mathcal{S} \times \mathcal{S}} \mu^\pi(s, s') \log \left( \frac{\mu^R(s, s')}{\mu^E(s, s')} \times \frac{\mu^\pi(s, s')}{\mu^R(s, s')} \right) ds ds' \\
&= \int_{\mathcal{S} \times \mathcal{S}} \mu^\pi(s, s') \log \frac{\mu^R(s, s')}{\mu^E(s, s')} ds ds' + \int_{\mathcal{S} \times \mathcal{A}} \mu^\pi(s, s') \log \frac{\mu^\pi(s, s')}{\mu^R(s, s')} ds ds' \\
&= \mathbb{E}_{\mu^\pi(s, s')}[\log \frac{\mu^R(s, s')}{\mu^E(s, s')}] + \mathbb{D}_{\mathbf{KL}}[\mu^\pi(s, s')||\mu^R(s, s')] \\
&\leq \mathbb{E}_{\mu^\pi(s, s')}[\log \frac{\mu^R(s, s')}{\mu^E(s, s')}] + \underbrace{\mathbb{D}_{\mathbf{KL}}[\mu^\pi(s, a)||\mu^R(s, a)]}_{\text{derived from Lemma 2}}.
\end{aligned}
$$

$\square$

## 9.2 Connections between LfO and LfD

**Theorem 2.**

$$\mathbb{D}_{\mathbf{KL}}[\mu^\pi(a|s, s')||\mu^E(a|s, s')] = \mathbb{D}_{\mathbf{KL}}[\mu^\pi(s, a)||\mu^E(s, a)] - \mathbb{D}_{\mathbf{KL}}[\mu^\pi(s, s')||\mu^E(s, s')].$$

*Proof.* We can refer Eq (12) from the proof of Lemma 2:

$$\mathbb{D}_{\mathbf{KL}}[\mu^\pi(s, a)||\mu^E(s, a)] = \mathbb{D}_{\mathbf{KL}}[\mu^\pi(s, s')||\mu^E(s, s')] + \mathbb{D}_{\mathbf{KL}}[\mu^\pi(a|s, s')||\mu^E(a|s, s')].$$

$\square$

## 9.3 An Unoptimizable Gap Between LfO and LfD

**Remark 1**: *In a non-injective MDP, the discrepancy of* $\mathbb{D}_{\mathbf{KL}}[\mu^\pi(a|s, s')||\mu^E(a|s, s')]$ *cannot be optimized without knowing expert actions.*

*Proof.* We provide proof with a counter-example. Consider a non-injective MDP in a tabular case, whose transition dynamics is shown in Table 3, with $|\mathcal{S}| = 3$, and $|\mathcal{A}| = 4$. Especially, there exists two actions which lead to the same deterministic transition, i.e. for $s_1, s_2 \in \mathcal{S}, \exists a_0, a_2 \in \mathcal{A}$, s.t. $P(s_2|s_1, a_2) = P(s_2|s_1, a_0) = 1$, as illustrated in Figure 5.

In this MDP, there is an *expert* policy $\pi_E$ as listed in Table 5. Trajectories generated by this expert are illustrated as blue lines in Figure 5. In a LfO scenario, a learning agent only has access to sequences

| $P$ | $a_0$ | $a_1$ | $a_2$ | $a_3$ |
|---|---|---|---|---|
| $P(s_1\|s_1,\cdot)$ | 0 | 1 | 0 | 0 |
| $P(s_2\|s_1,\cdot)$ | **1** | 0 | **1** | 0 |
| $P(s_3\|s_1,\cdot)$ | 0 | 0 | 0 | 1 |
| $P(s_1\|s_2,\cdot)$ | 0 | 1 | 0 | 0 |
| $P(s_2\|s_2,\cdot)$ | 0 | 0 | 1 | 0 |
| $P(s_3\|s_2,\cdot)$ | 0 | 0 | 0 | 1 |
| $P(s_1\|s_3,\cdot)$ | 0 | 1 | 0 | 0 |
| $P(s_2\|s_3,\cdot)$ | 0 | 0 | 1 | 0 |
| $P(s_3\|s_3,\cdot)$ | 0 | 0 | 0 | 1 |

Table 3: A deterministic but non-injective MDP.

| $\pi$ | $s_1$ | $s_2$ | $s_3$ |
|---|---|---|---|
| $a_0$ | **0.5** | 0 | 0 |
| $a_1$ | 0 | 0 | 1 |
| $a_2$ | **0.5** | 0 | 0 |
| $a_3$ | 0 | 1 | 0 |

Table 4: Learning Policy $\pi$.

| $\pi_E$ | $s_1$ | $s_2$ | $s_3$ |
|---|---|---|---|
| $a_0$ | 0 | 0 | 0 |
| $a_1$ | 0 | 0 | 1 |
| $a_2$ | **1** | 0 | 0 |
| $a_3$ | 0 | 1 | 0 |

Table 5: Expert Policy $\pi_E$.

Figure 5: Transition of an non-injective MDP.

of states visited by the expert: $\mathcal{R}_E = \{s_1, s_2, s_3, s_1, s_2, s_3, \cdots\}$, without knowing what actions have been taken by the expert.

Based on the given observations $\mathcal{R}_E$, a policy $\pi$ can only satisfy the state distribution matching with $\mathbb{D}_{\mathbf{KL}}[\mu^\pi(s, s')||\mu^E(s, s')] = 0$, but unable to optimize $\mathbb{D}_{\mathbf{KL}}[\mu^\pi(a|s, s)||\mu^E(a|s, s)]$, as both $a_0$ and $a_2$ lead to a deterministic transition of $s_1 \to s_2$. In lack of expert actions, the best guess for a learning policy is to equally distribute action probabilities with $\pi(a_0|s_1) = (a_2|s_1) = 0.5$. which results in $\mu^\pi(a_0|s_1, s_2) = \mu^\pi(a_2|s_1, s_2) = 0.5$, whereas $\mu^E(a_2|s_1, s_2) = 1$, $\mu^E(a_0|s_0, s_1) = 0$. Consequently, we reach at $\mathbb{D}_{\mathbf{KL}}[\mu^\pi(a|s, s')||\mu^E(a|s, s')] > 0$. $\square$

**Remark:** *In a deterministic and injective MDP, it satisfies that $\forall \pi : \mathcal{S} \to \mathcal{A}$, $\mathbb{D}_{\mathbf{KL}}[\mu^\pi(a|s, s')||\mu^E(a|s, s')] = 0$.*

We provide proof in a finite, **discrete** state-action space, although the conclusion is valid to extend to continuous cases.

*Proof.* In a deterministic and injective MDP, we can interpret the transition dynamics with a *deterministic* function g:

$\exists g : \mathcal{S} \times \mathcal{A} \to \mathcal{S}$, s.t. $\forall (s, a, s')$, $g(s, a) = s' \iff P(s'|s, a) = 1$, and $g(s, a) \neq s' \iff P(s'|s, a) = 0$.

since this MDP is also injective, given arbitrary policy $\pi$ and a transition $s \to s', (s, s') \sim \mu^\pi(s, s')$, there exists one and only action $a$ which satisfies $g(s, a) = s', P(s'|s, a) = 1$.

Accordingly, $\mu^\pi(a|s, s') = \frac{\pi(a|s)P(s'|s,a)}{\mathbb{E}_{\bar{a} \sim \pi(\cdot|s)}[P(s'|s,a)]} = \mathbb{1}[g(s, a) = s']$ depends only on the transition dynamics, where $\mathbb{1}(x)$ is an indicator function. The same conclusion applies to $\mu^E(a|s, s')$ as well. Therefore, we reach at:

$$\forall \pi : \mathcal{S} \to \mathcal{A}, \ \mathbb{D}_{\mathbf{KL}}[\mu^\pi(a|s, s')||\mu^E(a|s, s')]$$

$$= \mathbb{E}_{\mu^\pi(s,a,s')} \Big[ \log \frac{\mathbb{1}[g(s,a) = s']}{\mathbb{1}[g(s,a) = s']} \Big]$$

$$= \mathbb{E}_{\mu^\pi(s,a,s')} \Big[ \log \frac{1}{1} \Big] = 0.$$

$\square$

## 9.4 Upper-bound of the KL-Divergence

**Theorem 3.** *For two arbitrary distributions $P$ and $Q$, and an $f$-divergence with $f(x) = \frac{1}{2}x^2$, it satisfies that $\mathbb{D}_{\mathbf{KL}}[P||Q] \leq \mathbb{D}_f[P||Q]$.*

*Proof.* Given two distributions $P$ and $Q$, their density ratio is denoted as $w_{p|q}$, with $w_{p|q} = \frac{p(x)}{q(x)} \geq 0$. If we consider a function $g(w) = w\log(w) - \frac{1}{2}w^2$, $g(w)$ is constantly decreasing when $w \in (0,\infty)$, as $\frac{\partial g}{\partial w} = \log w + 1 - w \leq 0 \ \forall w \geq 0$.

Since KL-Divergence is a special case of $f$-divergence with $f_{\mathbf{KL}}(x) = x\log x$, it is sufficient to show that:

$$\mathbb{D}_{\mathbf{KL}}[P||Q] - \mathbb{D}_f[P||Q] = \int_{\mathcal{X}} q(x)\Big( w_{p/q}\log(w_{p/q}) - \frac{1}{2}(w_{p/q})^2 \Big) dx$$

$$\leq \int_{\mathcal{X}} q(x) \sup_{w \in (0,+\infty)} (w\log(w) - \frac{1}{2}w^2) dx$$

$$= \int_{\mathcal{X}} q(x) \lim_{w \to 0^+} (w\log(w) - \frac{1}{2}w^2) dx$$

$$= 0.$$

$\square$

## 9.5 Forward Distribution Matching

### 9.5.1 Lower-bound of the BC Objective

**Theorem 4.**

$$\mathbb{D}_{\mathbf{KL}}[\pi_E(a|s)||\pi(a|s)] = \mathbb{D}_{\mathbf{KL}}[\mu^E(s'|s)||\mu^\pi(s'|s)] + \mathbb{D}_{\mathbf{KL}}[\mu^E(a|s,s')||\mu^\pi(a|s,s')].$$

*Proof.* Based on the definition of $\mu^\pi(a|s,s')$ in Table 1:

$$\mu^\pi(a|s,s') = \frac{\pi(a|s)P(s'|s,a)}{\int_{\mathcal{A}} \pi(\bar{a}|s)P(s'|s,\bar{a})d\bar{a}} = \frac{\pi(a|s)P(s'|s,a)}{\mu^\pi(s'|s)}, \tag{13}$$

and similar for $\mu^E(a|s,s')$, we can derive at the following:

$$\mathbb{D}_{\mathbf{KL}}[\pi_E(a|s)||\pi(a|s)]$$

$$= \int_{\mathcal{S} \times \mathcal{A}} \mu^E(s)\pi_E(a|s) \log \frac{\pi_E(a|s)}{\pi(a|s)} dads$$

$$= \int_{\mathcal{S} \times \mathcal{A}} \mu^E(s,a) \log \frac{\pi_E(a|s)}{\pi(a|s)} dads$$

$$= \int_{\mathcal{S} \times \mathcal{A} \times \mathcal{S}} \mu^E(s,a)P(s'|s,a) \log \frac{\pi_E(a|s)P(s'|s,a)}{\pi(a|s)P(s'|s,a)} ds'dads$$

$$= \int_{\mathcal{S} \times \mathcal{A} \times \mathcal{S}} \mu^E(s,a,s') \log \frac{\pi_E(a|s)P(s'|s,a)}{\pi(a|s)P(s'|s,a)} ds'dads$$

$$= \int_{\mathcal{S} \times \mathcal{A} \times \mathcal{S}} \mu^E(s,a,s') \log \underbrace{\frac{\mu^E(a|s,s')\mu^E(s'|s)}{\mu^\pi(a|s,s')\mu^\pi(s'|s)}}_{\text{Eq (13)}} ds'dads$$

$$= \int_{\mathcal{S}\times\mathcal{A}\times\mathcal{S}} \mu^E(s,a,s')\Big(\log\frac{\mu^E(a|s,s')}{\mu^\pi(a|s,s')} + \log\frac{\mu^E(s'|s)}{\mu^\pi(s'|s)}\Big)ds'dads$$

$$= \int_{\mathcal{S}\times\mathcal{A}\times\mathcal{S}} \mu^E(s,a,s')\log\frac{\mu^E(a|s,s')}{\mu^\pi(a|s,s')}ds'dads + \int_{\mathcal{S}\times\mathcal{A}\times\mathcal{S}} \mu^E(s,a,s')\log\frac{\mu^E(s'|s)}{\mu^\pi(s'|s)}ds'dads$$

$$= \mathbb{D}_{\mathbf{KL}}[\mu^E(a|s,s')||\mu^\pi(a|s,s')] + \mathbb{D}_{\mathbf{KL}}[\mu^E(s'|s)||\mu^\pi(s'|s)].$$

$\square$

### 9.5.2 Policy Regularization as A Forward Distribution Matching

Without loss of generality, in this section we provide proof based on a finite, **discrete** state-action space.

**Assumption 1** (Deterministic MDP). $\exists g : \mathcal{S} \times \mathcal{A} \rightarrow \mathcal{S}$ *a deterministic function, s.t.* $\forall (s,a,s')$, $g(s,a)\neq s' \iff P(s'|s,a)=0$, *and* $g(s,a)=s' \iff P(s'|s,a)=1$.

Based on Assumption 1, we have the following:

**Corollary 1.** *In a deterministic MDP,* $\forall \pi : \mathcal{S}\rightarrow\mathcal{A}$, $\mu^\pi(a|s,s')>0 \Longrightarrow P(a|s,s')=1$.

*Proof.* $\mu^\pi(a|s,s') \propto \pi(a|s)P(s'|s,a) > 0 \Longrightarrow P(s'|s,a) > 0$. Based on Assumption 1, it holds that $g(s,a) = s'$, therefore $P(s'|s,a) = 1$. $\square$

**Assumption 2** (Support Coverage). *The support of expert transition distribution* $\mu^E(s,s')$ *is covered by* $\mu^R(s,s')$:

$$\mu^E(s,s') > 0 \Longrightarrow \mu^R(s,s') > 0.$$

Combing Corollary 1 and Assumption 2, we can reach at the following:

**Corollary 2.** $\forall(s,s')\sim\mu^E(s,s'), \mu^R(a|s,s')>0 \Longrightarrow P(a|s,s')=1$.

**Lemma 3.** *Given a policy* $\hat{\pi}$, *s.t.* $\forall(s,s')\sim\mu^E(s,s')$, $\hat{\pi}(a|s)\propto\mu^R(a|s,s')$, *then it satisfies that:*

$$\forall\pi:\mathcal{S}\rightarrow\mathcal{A},\ \mathbb{D}_{\mathbf{KL}}[\mu^E(s'|s)||\mu^\pi(s'|s)] \geq \mathbb{D}_{\mathbf{KL}}[\mu^E(s'|s)||\mu^{\hat{\pi}}(s'|s)].$$

*Proof.* In a discrete state-action space, $\mu^\pi(s'|s)$ can be denoted as $\mu^\pi(s'|s) = \mathbb{E}_{a\sim\pi(\cdot|s)}[P(s'|s,a)]$, and the similar for $\mu^{\hat{\pi}}(s'|s)$:

$$\mathbb{D}_{\mathbf{KL}}[\mu^E(s'|s)||\mu^{\hat{\pi}}(s'|s)] - \mathbb{D}_{\mathbf{KL}}[\mu^E(s'|s)||\mu^\pi(s'|s)]$$

$$=\mathbb{E}_{\mu^E(s,s')}\left[\log\frac{\mu^E(s'|s)}{\mu^{\hat{\pi}}(s'|s)} - \log\frac{\mu^E(s'|s)}{\mu^\pi(s'|s)}\right]$$

$$=\mathbb{E}_{\mu^E(s,s')}\left[\log\mu^\pi(s'|s) - \log\mu^{\hat{\pi}}(s'|s)\right]$$

$$=\mathbb{E}_{\mu^E(s,s')}\left[\log\mathbb{E}_{a\sim\pi(\cdot|s)}[P(s'|s,a)]\right] - \mathbb{E}_{\mu^E(s,s')}\left[\log\mathbb{E}_{a\sim\hat{\pi}(\cdot|s)}[P(s'|s,a)]\right]$$

$$=\mathbb{E}_{\mu^E(s,s')}\left[\log\mathbb{E}_{a\sim\pi(\cdot|s)}[P(s'|s,a)]\right] - \mathbb{E}_{\mu^E(s,s')}\left[\log\mathbb{E}_{a\sim\mu^R(\cdot|s,s')}[P(s'|s,a)]\right]$$

$$=\mathbb{E}_{\mu^E(s,s')}\left[\log\mathbb{E}_{a\sim\pi(\cdot|s)}[P(s'|s,a)]\right] - \mathbb{E}_{\mu^E(s,s')}\underbrace{\left[\log\mathbb{E}_{a\sim\mu^R(\cdot|s,s')}[1]\right]}_{\text{Corollary 2}}$$

$$=\mathbb{E}_{\mu^E(s,s')}\left[\log\mathbb{E}_{a\sim\pi(\cdot|s)}[P(s'|s,a)]\right]$$

$$\leq\mathbb{E}_{\mu^E(s,s')}\left[\log\mathbb{E}_{a\sim\pi(\cdot|s)}[1]\right]$$

$$=0.$$

$\square$

**Remark 2.** *In a deterministic MDP, assuming the support of* $\mu^E(s,s')$ *is covered by* $\mu^R(s,s)$, *s.t.* $\mu^E(s,s') > 0 \implies \mu^R(s,s') > 0$, *then regulating policy using* $\mu^R(\cdot|s,s')$ *can minimize* $\mathbb{D}_{\mathbf{KL}}[\mu^E(s'|s)||\mu^\pi(s'|s)]$:

$$\exists\tilde{\pi}:\mathcal{S}\rightarrow\mathcal{A},\ s.t.\ \forall(s,s')\sim\mu^E(s,s'),\ \tilde{\pi}(\cdot|s)\propto\mu^R(\cdot|s,s') \Longrightarrow \tilde{\pi} = \arg\min_\pi\mathbb{D}_{\mathbf{KL}}[\mu^E(s'|s)||\mu^\pi(s'|s)].$$

*Proof.* Based on Lemma 3, we have that:
$$\forall \pi : \mathcal{S} \to \mathcal{A}, \ \mathbb{D}_{\mathbf{KL}}[\mu^E(s'|s)||\mu^{\pi}(s'|s)] \geq \mathbb{D}_{\mathbf{KL}}[\mu^E(s'|s)||\mu^{\tilde{\pi}}(s'|s)].$$
Therefore, $\tilde{\pi} = \arg\min_{\pi} \mathbb{D}_{\mathbf{KL}}[\mu^E(s'|s)||\mu^{\pi}(s'|s)].$

□

### 9.5.3 Estimating the Inverse Action Distribution

**Theorem 5.**
$$\max_{P_I:\mathcal{S}\times\mathcal{S}\to\mathcal{A}} -\mathbb{D}_{\mathbf{KL}}[\mu^R(a|s,s')||P_I(a|s,s')] \equiv \max_{P_I:\mathcal{S}\times\mathcal{S}\to\mathcal{A}} \mathbb{E}_{(s,a,s')\sim\mu^R(s,a,s')}[\log P_I(a|s,s')].$$

*Proof.*
$$- \mathbb{D}_{\mathbf{KL}}[\mu^R(a|s,s')||P_I(a|s,s')]$$
$$= -\int_{\mathcal{S}\times\mathcal{S}\times\mathcal{A}} \mu^R(s,s')\mu^R(a|s,s') \log \frac{\mu^R(a|s,s')}{P_I(a|s,s')} dadsds'$$
$$= -\int_{\mathcal{S}\times\mathcal{S}\times\mathcal{A}} \mu^R(s,s')\mu^R(a|s,s') \Big( \log \mu^R(a|s,s') - \log P_I(a|s,s') \Big) dadsds'$$
$$= \underbrace{H[\mu^R(a|s,s')]}_{\text{fixed w.r.t. } P_I} + \int_{\mathcal{S}\times\mathcal{S}\times\mathcal{A}} \mu^R(s,s')\mu^R(a|s,s') \log P_I(a|s,s') dadsds'$$
$$= \underbrace{H[\mu^R(a|s,s')]}_{\text{fixed w.r.t. } P_I} + \mathbb{E}_{\mu^R(s,a,s')}[\log P_I(a|s,s')].$$

□

Note that we use $H[\mu^R(a|s,s')]$ to denote the conditional entropy of $\mu^R(a|s,s')$, with $H[\mu^R(a|s,s')] = \mathbb{E}_{\mu^R(s,a,s')}[-\log \mu^R(a|s,s')]$.

## 9.6 Derivation of Eq (8):

$$J_{\text{opolo}}(\pi, Q) = \mathbb{E}_{(s,a,s')\sim\mu^{\pi}(s,a,s')}[r(s,s') - (\mathcal{B}^{\pi}Q - Q)(s,a)] + \mathbb{E}_{(s,a)\sim\mu^R(s,a)}[f_*((\mathcal{B}^{\pi}Q - Q)(s,a))],$$
where $\mathcal{B}^{\pi}Q(s,a) = \mathbb{E}_{s'\sim P(\cdot|s,a),a'\sim\pi(\cdot|s')}\Big[r(s,s') + \gamma Q(s',a')\Big]$, and $r(s,s') = \log \frac{\mu^E(s,s')}{\mu^R(s,s')}$.

*Proof.* The first term in the RHS of the above equation can be reduced to the following:
$$\mathbb{E}_{(s,a,s')\sim\mu^{\pi}(s,a,s')}[r(s,s') - (\mathcal{B}^{\pi}Q - Q)(s,a)]$$
$$= \mathbb{E}_{(s,a)\sim\mu^{\pi}(s,a)}\Big[\mathbb{E}_{s'\sim P(\cdot|s,a)}[r(s,s') - ((\mathcal{B}^{\pi}Q - Q)(s,a))]\Big]$$
$$= \mathbb{E}_{(s,a)\sim\mu^{\pi}(s,a)}\Big[\mathbb{E}_{s'\sim P(\cdot|s,a)}[r(s,s')] + Q(s,a) - \mathbb{E}_{s'\sim P(\cdot|s,a)}[\mathcal{B}^{\pi}Q(s,a)]\Big]$$
$$= \mathbb{E}_{(s,a)\sim\mu^{\pi}(s,a)}\Big[\mathbb{E}_{s'\sim P(\cdot|s,a)}[\cancel{r(s,s')}] + Q(s,a) - \mathbb{E}_{s'\sim P(\cdot|s,a),a'\sim\pi(\cdot|s')}[\cancel{r(s,s')} + \gamma Q(s',a')]\Big]$$
$$= \mathbb{E}_{(s,a)\sim\mu^{\pi}(s,a)}\Big[Q(s,a) - \gamma\mathbb{E}_{s'\sim P(\cdot|s,a),a'\sim\pi(\cdot|s')}[Q(s',a')]\Big]$$
$$= \underbrace{(1-\gamma)\sum_{t=0}^{\infty}\gamma^t\mathbb{E}_{s\sim\mu_t^{\pi}(s),a\sim\pi(s)}[Q(s,a)]}_{\text{see Table 1}} - (1-\gamma)\sum_{t=0}^{\infty}\gamma^{t+1}\mathbb{E}_{s\sim\mu_t^{\pi},a\sim\pi(\cdot|s),s'\sim P(\cdot|s,a),a'\sim\pi(\cdot|s')}[Q(s',a')]]$$
$$= (1-\gamma)\sum_{t=0}^{\infty}\gamma^t\mathbb{E}_{s\sim\mu_t^{\pi},a\sim\pi(s)}[Q(s,a)] - (1-\gamma)\sum_{t=0}^{\infty}\gamma^{t+1}\mathbb{E}_{s\sim\mu_{t+1}^{\pi},a\sim\pi(\cdot|s)}[Q(s,a)]]$$
$$= (1-\gamma)\mathbb{E}_{s\sim p_0,a_0\sim\pi(\cdot|s_0)}[Q(s_0,a_0)].$$
Therefore:
$$J_{\text{opolo}}(\pi, Q) = (1-\gamma)\mathbb{E}_{s\sim p_0,a_0\sim\pi(\cdot|s_0)}[Q(s_0,a_0)] + E_{(s,a)\sim\mu^R}[f_*((\mathcal{B}^{\pi}Q - Q)(s,a))].$$

□

### 9.7 Implementation Details

#### 9.7.1 Practical Considerations for Algorithm Implementation

We provide some practical considerations to effectively implement our algorithm:

**Initial state sampling:** To increase the diversity of initial samples, we use state samples from an off-policy buffer and treat them as *virtual initial states*. A similar strategy is adopted by [3].

**Constant shift on synthetic rewards**: In practice, we adopt the same strategy of prior art [10] to use $r(s, s') = -\log(1 - D(s, s'))$, instead of $\log(D) - \log(1 - D)$ as the discriminator output. A fully optimized discriminator $D^*$ satisfies $-\log(1 - D^*(s, s')) = \log(1 + \frac{\mu^E(s,s')}{\mu^R(s,s')})$, which corresponds to a constant shift on $\frac{\mu^E(s,s')}{\mu^R(s,s')}$ before the log term.

**Q and $\pi$ network update:** We follow the advice of AlgeaDICE [31] by using a target Q network and policy gradient clipping. Especially, when taking the gradients of $J_{\text{opolo}}(\pi, Q, \alpha)$ w.r.t.$Q$, we use the value from a target Q network to calculate $\mathcal{B}^\pi Q(s, a)$ in order to stabilize training; on the other hand, since an optimal $x^*(s, a) = (\mathcal{B}^\pi Q^* - Q^*)(s, a) = \frac{\mu^\pi(s,a)}{\mu^R(s,a)}$ represents a density ratio and should always be non-negative, we clip $(\mathcal{B}^\pi Q - Q)(s, a)$ to above 0 when taking gradients w.r.t.$\pi$.

#### 9.7.2 Hyper-parameters

Table 6 lists the hyper-parameters for GAIL [2], GAIfO [9], BCO [17], DAC [4], and our proposed approach *OPOLO*. Specifically, for off-policy approaches, each self-generated interaction will be stored the replay buffer in a FIFO manner, and *update frequency* is the number of interactions sampled from the MDP after which the module is updated. Moreover, considering the different scales for the gradients of $J(\pi_\theta, Q_\phi)$ and $J_{\text{Reg}}(\pi_\theta)$ in Algorithm 1, we apply a coefficient $\lambda$ for *OPOLO* to adjust the regularization strength when calculating the total policy loss:

$$\theta \leftarrow \theta + \alpha\big(J_{\nabla\theta}(\pi_\theta, Q_\phi) + \lambda J_{\nabla\theta} J_{\text{Reg}}(\pi_\theta)\big).$$

| Hyper-parameters | Value |
|---|---|
| **Shared Parameters for Off-Policy Approaches** | |
| Buffer size | $10^7$ |
| Batch size | 100 |
| Learning rate | $3e^{-4}$ |
| Discount factor $\gamma$ | 0.99 |
| Network architecture | MLP [400, 300] |
| $Q, \pi$ update frequency / gradient steps | $10^3/10^3$ |
| $D$ update frequency / gradient steps | $500/10$ |
| **Shared Parameters for On-Policy Approaches** | |
| Batch size | 2048 |
| mini-Batch size | 256 |
| Learning rate | $3e^{-4}$ |
| Discount factor $\gamma$ | 0.99 |
| Network architecture | MLP [400, 300] |
| **BCO** | |
| $P_I$ pre-train gradient steps | $10^4$ |
| $P_I$ update frequency / gradient steps | $10^3/100$ |
| **DAC** | |
| Number of extra absorbing states | 1 |
| ***OPOLO*** | |
| $P_I$ update frequency / gradient steps | $500/50$ |
| $P_I$ regularization coefficient $\lambda$ | 0.1 |

Table 6: Hyper-parameters for Different Algorithms

## 9.8 Challenges of DICE without Expert Actions

In this section, we analyze the principle of offline imitation learning using DICE [30, 33, 31] and the reason that impedes its direct application to an LfO setting.

In a LfO setting where expert actions are unavailable, the learning objective is to minimize the discrepancy of *state-only* distributions induced by the agent and the expert. Without loss of generality, we consider an arbitrary f-divergence $\mathbb{D}_f$ as the discrepancy measure:

$$\max_\pi -\mathbb{D}_f[\mu^\pi(s,s')||\mu^E(s,s')]$$

$$= \max_\pi \min_{x:\mathcal{S}\times\mathcal{S}\to\mathbb{R}} \mathbb{E}_{\mu^\pi(s,s')}[-x(s,s')] + \mathbb{E}_{\mu^E(s,s')}[f^*(x(s,s'))], \tag{14}$$

in which $f^*(x)$ is the conjugate of $f(x)$ for the $f$-divergence. To remove the on-policy dependence of $\mu^\pi(s,s')$, we follow the rationale of DICE and use a similar change-of-variable trick mentioned in Sec 3.2 to learn a value function $v(s,s')$:

$$v(s,s') := -x(s,s') + \gamma\mathbb{E}_{a'\sim\pi(.|s'),s''\sim P(.|s',a')}[v(s',s'')] = -x(s,s') + \mathcal{B}^\pi v(s,s').$$

This value function is a fixed point solution to an variant Bellman operator $\mathcal{B}^\pi$, which, however, is problematic in a model-free setting. To see this, we substitute $x(s,s')$ by $(\mathcal{B}^\pi v - v)(s,s')$ to transform Eq (14) into the following:

$$\max_\pi \min_{x:\mathcal{S}\times\mathcal{S}\to\mathbb{R}} \mathbb{E}_{\mu^\pi(s,s')}[-x(s,s')] + E_{\mu^E(s,s')}[f^*(x(s,s'))]$$

$$= \max_\pi \min_{v:\mathcal{S}\times\mathcal{S}\to\mathbb{R}} (1-\gamma)\underbrace{\mathbb{E}_{s_0\sim p_0,s_1\sim P(\cdot|s_0,\pi(s_0))}[v(s_0,s_1)]}_{\text{term 1}} + \underbrace{\mathbb{E}_{\mu^E(s,s')}[f^*((\mathcal{B}^\pi v - v)(s,s'))]}_{\text{term 2}}.$$

where $\mathcal{B}^\pi v(s,s') = \gamma\mathbb{E}_{a'\sim\pi(.|s'),s''\sim P(.|s',a')}[v(s',s'')]$. Optimizing this objective is troublesome, in that the $\mathcal{B}^\pi v(s,s')$ in term 2 requires knowledge of $P(\cdot|s,\pi(s))$, $\forall s\sim\mu^E(s)$. In another word, for any state sampled from the *expert* distribution, we need to know what would be the *next* state if following policy $\pi$ from this state. A similar issue is echoed in term 1, where $s_1$ is sampled from $P(\cdot|s_0,\pi(s_0))$. Consequently, directly applying DICE loses its advantage in a LfO setting, as it incurs a dependence on a *forward transition* model, which is costly to estimate and may counteract the efficiency brought by off-policy learning.