[Reviews · NeurIPS 2020]

Review 1

Summary and Contributions: The paper proposes an off-policy algorithm for imitation learning from observation called OPOLO to bring down the number of required environment interactions to learn tasks with adversarial imitation from observation algorithms. The way that it works is that it learns a critic along with the discriminator and the policy. The paper proposes an inverse action regularizer as well which has shown in the experiments that it helps with the number of environment interactions without losing much of the performance.

Strengths: One challenge in adversarial imitation learning algorithms is that they require a lot of environment interactions to learn a task. So making the algorithms off-policy, would result in potentially implementing it in the real world. The algorithm is tested in 6 MuJoCo domains which shows comparable final performance with a lot of improvements in the number of interaction steps by a large margin compared to GAIfO which is the most related to OPOLO. Also the results show that inverse action regularizer helps with the number of interaction step without losing much of the performance. The algorithm is novel and as it is about imitation learning, it is relevant to the NeurIPS community.

Weaknesses: The experiment domains are relatively simple. More complex domains would have given better insight on how the algorithm works.

Correctness: The claims and the method seem to be correct and the empirical methodology and the experiments are sound.

Clarity: The paper is well-written and easy to follow.

Relation to Prior Work: The paper has properly cited previous works and described how the work is related to them.

Reproducibility: Yes

Additional Feedback:


Review 2

Summary and Contributions: The paper presents an off-policy approach to imitation learning from observations. Starting with the KL minimization objective, an upper bound is created using an f-divergence, which is transformed into a min-max saddle-point optimization with the use of Fenchel conjugates. The dependence on on-policy samples is removed using the change-of-variables trick from DualDICE. Furthermore, the policy is regularized by maximizing the likelihood of action generated from a learned inverse action model on the state-transitions observed by the expert. ----- Post-rebuttal update ------ As per suggestion, the authors have included two new baselines (DACfO and ValueDICEfO). Although the empirical advantage of the proposed method over the best baseline, averaged over the environments, is somewhat reduced, I believe there's enough merit in the paper to recommend acceptance. I have updated my rating accordingly.

Strengths: Tackles a pertinent problem of imitation only from observations with an off-policy RL algorithm. Clear presentation and sound claims.

Weaknesses: Comparison to stronger baselines. Please see "Additional feedback, comments, suggestions for improvement and questions for the authors" for details.

Correctness: Yes.

Clarity: Yes.

Relation to Prior Work: Yes.

Reproducibility: Yes

Additional Feedback: As much as I like the presentation of the paper, it is worth noting that there are significant “conceptual” similarities with prior work – 1.) making the objective off-policy (which arguably is the main point of the paper) is a direct application of the change-of-variables trick was proposed in [1] and most recently used in [2] for imitation learning from observations and actions; 2.) The regularization using the inverse model appears in [3] though the direction of KL is different and [3] uses a VAE to predict the expert-like next-states. My main concern though is the lack of representative off-policy LfO baselines – the authors use BCO for this which is quite weak. It would be useful to compare with the following: 1. Off-policy LfO version of DAC. DAC uses off-policy (TD3) for policy learning and simply ignores the importance-correction in the discriminator training. This baseline could be used by replacing (s, a) with (s,s’) in the discriminator, keeping other pieces the same. 2. [2] with learned inverse action model. It seems the inverse action model works quite well for the environments considered in this paper. Could [2] be used as a baseline by providing ‘pseudo’ expert actions from the learned inverse action model? I am flexible with my rating if the authors could include either (or both) baseline, or alternatively argue why these baselines are not suitable for LfO and/or wouldn’t be performant. Notation – in Supplementary 8.5.2, at a couple of places it say P(a|s,s’) = 1. Should that be P(s’|s,a) = 1? [1] dualdice: behavior-agnostic estimation of discounted stationary distribution corrections. [2] imitation Learning via Off-Policy Distribution Matching [3] state alignment-based imitation learning


Review 3

Summary and Contributions: The paper provides a new off-policy imitation learning algorithm where only expert observations are available. The method builds upon recent advances in imitation learning with f-divergence and distribution error correction in off-policy reinforcement learning.

Strengths: The authors provided detailed derivation of their method with theoretical justification of each component of the algorithm. Empirically, with very limited expert observations, the proposed method achieves competitive performance comparing with other off-policy imitation learning algorithms (some even with access to expert action).

Weaknesses: 1) It could be better if a more comprehensive comparison of the asymptotic performances among on/off-policy methods. Off policy methods enjoy better sample efficiency at the cost of higher computation burden. Maybe an additional table could be provided in the appendix. Personally, I don’t actually expect a gap between the on/off-policy methods. 2) Although the usage of current f-function in the f-divergence is justified, it will still be interesting to see a comparison if the alternative f-functions were adopted. An empirical comparison would further justify the usage of the author’s current choice. Minor: 1) In eq. 12 (the proof of Lemma 2), in the third from the last line, the integration should be over \mathcal{S} \times \mathcal{S} instead of \mathcal{S} \times \mathcal{A} \times \mathcal{S}. 2) Notations in sec. 8.6 are inconsistent: sampling from \pi is switching back and forth between $a \sim \pi(s)$ and $a \sim \pi(\cdot | s)$. 3) At the end of Algorithm box 1, does J_{\nabla \theta} J_{reg} (\pi_{\theta}) intends to mean that the gradient of \theta on J_{reg}? 4) Can the authors slightly justify the first equation in sec. 3.2? Why is that an equality instead of an inequality?

Correctness: Yes.

Clarity: Yes.

Relation to Prior Work: The choice that using the dual form of f-divergence seems also been raised in previous work in imitation learning: Ke L, Barnes M, Sun W, Lee G, Choudhury S, Srinivasa S. Imitation Learning as f-Divergence Minimization

Reproducibility: Yes

Additional Feedback:


Review 4

Summary and Contributions: The paper an off-policy algorithm (an extension of ValueDICE [3]) for imitating trajectories without known actions. The method is a clever adaptation of adversarial imitation learning for f-divergence minimization. While recent off-policy work trained the adversarial network on the replay buffer instead of the policy distribution, the paper leverages a different f-divergence (through the dual form) and a telescoping trick similar to DICE to obtain an off-policy objective that minimizes precisely the divergence between the transitions under current policy and the demonstration transitions. An additional behavior cloning regularization is used. This produces a more data-efficient algorithm due to use of off-policy data, while the algorithm moreover does not require the expert actions to be known. It is shown that the method outperforms prior work in terms of data efficiency and sometimes final performance on a set of MuJoCo tasks, this including some prior work that has access to expert actions. ---- Decision ---- The paper proposes a principled method with good performance. The techniques used in the derivation of the objective (while largely borrowed from [3]) are likely to be impactful for development of future methods, and the empirical performance of the method suggests it maybe become widely used for learning from observations with unknown actions. While the paper would be much improved with better baselines and harder environments, these issues are not critical for publication and may be resolved in the rebuttal. I therefore recommend acceptance. ---- Update ---- The authors made a significant attempt to improve the paper in the rebuttal, addressing most of my concerns. I maintain my acceptance rating. Additional comments: One finding that I think is quite curious is that OPOLO outperforms ValueDICE. This is somewhat surprising, since the algorithms are very similar, with OPOLO being provided less information (i.e. no actions). This suggests that the proposed method in fact may have broader than expected impact. Perhaps the authors could discuss this in the final version.

Strengths: The paper proposes a principled and data-efficient method for off-policy learning from demonstrations without actions. Furthermore, the method outperforms several alternatives, including a prior method that has access to actions, and recovers expert performance, proving the value of the method for future practitioners.

Weaknesses: While the paper is generally very solid, it could be much improved with more solid baseline comparisons. The comparison to DAC is nice as it shows that the proposed method is competitive even to some methods that use action data, but a comparison to a state-of-the-art method such as ValueDICE would be more appropriate. Furthermore, while the paper compares to a method that uses an inverse model for imitation without actions, comparison to methods that use forward models such as [39] are missing. Moreover, it would be interesting to see harder environments such as the ones in the Adroit suite (Rajeswaran'17). Rajeswaran'19, Learning Complex Dexterous Manipulation with Deep Reinforcement Learning and Demonstrations

Correctness: Yes.

Clarity: The paper is well-written and the technical exposition is very clear. A minor issue is that Figure 1 is being referred to before Table 2 but appears after it.

Relation to Prior Work: The paper has a good overview of related work, but misses some of the more recent papers like Edwards'20, Schmeckpeper'20. The paper would also benefit from better baselines, like [3] and [39]. Edwards'20, Estimating Q(s, s') with Deep Deterministic Dynamics Gradients Schmeckpeper'20, Learning Predictive Models From Observation and Interaction

Reproducibility: Yes

Additional Feedback: A minor note is that the paper does not in fact evaluate how critical the 'staleness' of the discriminator is. While in DAC the discriminator would indeed be trained on stale data, I suspect this might still correspond to a similar objective. In fact, such staleness is commonly induced _on purpose_ for stabilizing adversarial training both in computer vision and reinforcement learning. To fully support the claim that OPOLO is preferable because it does not suffer from the staleness issue, it would be good to analyze this experimentally, e.g. by training the DAC discriminator on only newer data.

[Author Response · NeurIPS 2020]

We thank all reviewers for their valuable feedback. We are encouraged that they found our algorithm novel (**R1**), our
paper well-written (**R1**, **R2**, **R3**, **R4**) with sound claims (**R2**), solid theoretical justifications (**R3**), and clear technical
expositions (**R4**). We are honored that **R4** recognizes the potential value of our work to the RL community. We provide
detailed responses to their major concerns below:

[**R1**, **R4**]: **1. Evaluation on more complex domains.** We appreciate this valuable suggestion. To better illustrate the
performance of our approach, we provide more evaluations on the *Humanoid* task (given the limited time constraint),
which is a challenging domain with high state-action dimension ($\mathcal{S} \times \mathcal{A} = \mathbb{R}^{376} \times \mathbb{R}^{17}$). The strength of *OPOLO* is
more significant in this domain ((Figure 1), while its counterparts can be prone to sub-optimality (*DAC*) or overfitting
(*ValueDICEfO*) (see our response **2**).

[**R2**, **R4**]: **2. Solid comparison with stronger baselines.** Following this informative suggestion, we compare with
three more baselines: ① *ValueDICE* as **R4** mentioned. We would like to emphasize that *ValueDICE* is a LfD approach
which is ***not directly applicable*** to LfO (see Sec 8.8), as it requires the expert actions at our disposal. For fairer
comparisons, we implemented its variant ② *ValueDICEfO* (as suggested by **R2**), which replaces ground-truth expert
actions with pseudo ones provided by an inverse model. Thanks to **R2**'s valuable suggestions, we also implemented
③ *DACfO*, a variation of *DAC* that learns the discriminator on $(s, s')$ instead of $(s, a)$; Although *ValueDICEfO* and
*DACfO* ***have not been investigated by other prior work***, we still found them quite interesting and relevant to our setting.
**Results** in Figure 1 (learning efficiency) and Table 2 (asymptotic performance) shows that: *OPOLO* (blue) in general 1)
learns ***faster*** than *DACfO* (red), 2) yields ***higher*** asymptotic performance than *DACfO* and *ValueDICEfO* (green), and
3) is more ***robust*** than other off-policy baselines including *ValueDICE* (orange) which uses expert actions. *OPOLO* is
the only approach that consistently achieves competitive performance regarding both sample-efficiency and asymptotic
performance across all tasks, and is therefore more stable compared with *ValueDICE*. As for the LfO baseline
*ValueDICEfO*, its performance compared with *ValueDICE* can be further deteriorated by potential *action-drifts*, as the
inferred actions are not guaranteed to recover expertise (see Sec 3.4 and Sec 8.3).

[**R3**]: **3. Comparison with other choices of $f$-divergence.** Following this valuable suggestion, we evaluated the
effects of different $f$-functions, where $f(x) = \frac{1}{p}|x|^p, f^*(y) = \frac{1}{q}|y|^q$, s.t. $\frac{1}{p} + \frac{1}{q} = 1, p, q > 1$, as adopted by *DualDICE*
(Nachum'19). We observed that *OPOLO* yields reasonable performance across different $f$-functions, although our
choice ($q = p = 2$) turns out to be most stable. Results using the *Ant* task is illustrated in Figure 2.

[**R2**]: **4. Conceptual resemblance to prior art: *DICE* and the inverse-action regularization.** We appreciate this
insightful comment for drawing a nice connection between *OPOLO* and other prior arts. We would like to highlight that:
1) Our approach is inspired by while different from *DICE* , as it is the first work to extend *DICE* to a more challenging
scenario (LfO), which is non-trivial especially when the philosophy of *DICE* is not directly applicable to this setting,
for which we have provided theoretical analysis (Sec 8.8). 2) Unlike prior art that empirically validated the effects of an
inverse-action model, we provide solid interpretations of its functionality, i.e. a *mode-covering* regularizer, by both
theoretical derivations and empirical ablation studies.

[**R4**]: **5. Effects of learning discriminator using fresh data.** We appreciate this insightful suggestion. We had similar
ideas before, by training discriminator $D$ using *on-policy* data, which did not bring us much benefit in terms of the
learning efficiency. We attribute this phenomenon to a *training distribution drift*, i.e. the *on-policy* dataset seen by $D$
differs from the *off-policy* ones used to train $\pi$ and $Q$, and the (potential) overfitting of $D$ may cause it forget on how to
distinguish stale (off-policy) samples. We consider it analogous to a *catastrophic forgetting* issue.

| Env | HalfCheetah | Hopper | Walker | Swimmer | Ant | Humanoid |
|---|---|---|---|---|---|---|
| *opolo*(-x) | **7632.80±128.88** | 3581.85±19.08 | 3947.72±97.88 | 257.38±4.28 | **5783.57±651.98** | **4699.68±1245.81** |
| *DAC* | 6900.00±131.24 | 3534.42±10.27 | **4131.05±174.13** | 232.12±2.04 | 5424.28±594.82 | 2303.97±379.28 |
| *DACfO* | 7035.63±444.14 | 3522.95±93.15 | 3033.02±207.63 | 185.28±2.67 | 4920.76±872.66 | 640.49±233.43 |
| *ValueDICE* | 5696.94±2116.94 | **3591.37±8.60** | 1641.58±1230.73 | **262.73±7.76** | 3486.87±1232.25 | 942.47±730.13 |
| *ValueDICEfO* | 4770.37±644.49 | 3579.51±10.23 | 431.00±140.87 | **265.05±3.45** | 75.08±400.87 | 198.39±65.46 |

Table 1: Performance after training with $10^6$ interaction steps

Figure 1: Learning curves averaged over 3 random seeds.

Figure 2: different $f$-functions.

[Meta-Review · NeurIPS 2020]

Reviewers agreed that this paper makes a good contribution, and enjoyed the principled derivation of the algorithm building upon the f-divergence for imitation learning and the distribution error correction in off-policy RL.